# The Intriguing Mystery of RPA Phosphorylation in DNA Double-Strand Break Repair

**DOI:** 10.3390/genes15020167

**Published:** 2024-01-27

**Authors:** Valerie J. Fousek-Schuller, Gloria E. O. Borgstahl

**Affiliations:** 1Immunology, Pathology, and Infectious Diseases, UNMC, Omaha, NE 68198-6805, USA; valerie.fousek@unmc.edu; 2Eppley Institute for Research in Cancer & Allied Diseases, UNMC, Omaha, NE 68198-6805, USA

**Keywords:** Replication Protein A (RPA), phosphorylation, homologous recombination, AlphaFold, protein-ssDNA interactions, cell cycle, DNA metabolism, double-strand break repair

## Abstract

Human Replication Protein A (RPA) was historically discovered as one of the six components needed to reconstitute simian virus 40 DNA replication from purified components. RPA is now known to be involved in all DNA metabolism pathways that involve single-stranded DNA (ssDNA). Heterotrimeric RPA comprises several domains connected by flexible linkers and is heavily regulated by post-translational modifications (PTMs). The structure of RPA has been challenging to obtain. Various structural methods have been applied, but a complete understanding of RPA’s flexible structure, its function, and how it is regulated by PTMs has yet to be obtained. This review will summarize recent literature concerning how RPA is phosphorylated in the cell cycle, the structural analysis of RPA, DNA and protein interactions involving RPA, and how PTMs regulate RPA activity and complex formation in double-strand break repair. There are many holes in our understanding of this research area. We will conclude with perspectives for future research on how RPA PTMs control double-strand break repair in the cell cycle.

## 1. Introduction

Replication protein A (RPA) is a multifaceted, heterotrimeric protein involved in various DNA metabolism activities, including cell cycle regulation, DNA damage signaling, and DNA repair mechanisms [1]. RPA is the main ssDNA-binding protein in humans. Historically, RPA was first discovered in the Kelly laboratory at Johns Hopkins University and identified as one of six purified components needed for in vitro simian virus 40 DNA replication. Then, RPA was found to play essential roles in all DNA metabolism pathways that involve single-stranded DNA (ssDNA) [2]. RPA research was pioneered and studied throughout Dr. Marc Wold’s entire career, who discovered the critical role of RPA in various pathways and the importance of RPA regulation by post-translational modifications (PTMs) [3,4,5,6,7,8,9,10,11,12,13,14,15,16,17,18,19,20,21,22,23,24,25,26,27,28,29,30,31,32,33]. RPA research has dramatically expanded and is of interest to many scientists. The RPA heterotrimer has three subunits named after their corresponding molecular weights: RPA70, RPA32, and RPA14 [34].

Over the years, research has shown that RPA has multiple types and patterns of PTMs, including acetylation, ubiquitination, SUMOylation, ADP ribosylation, and phosphorylation [28,35,36,37,38]. In this review, we focus on the role of phosphorylation. Many studies dating back to 1990 have focused on the phosphorylation of RPA, specifically the N-terminal domain of RPA32, because of the impact phosphorylation has on cell cycle regulation [39]. Also, the phosphorylation of RPA32 is easy to detect due to a noticeable shift to a higher molecular weight of the RPA32 subunit on SDS-PAGE. RPA32 has a long, unstructured N-terminal tail that is phosphorylated, and these sites have several phosphospecific antibodies. The phosphorylation of other subunits has also been studied through western blot and isoelectric focusing analysis of phosphoisoforms [40,41]. However, reagents for these sites could be more developed.

Since RPA phosphorylation before and after DNA damage has been shown, it is important to determine which sites on RPA are most important for the function of RPA to repair DNA damage, such as double-stranded breaks (DSBs) [40]. Previous data revealed that phosphatidyl inositol 3-kinase-like serine/threonine protein kinase (PIKK) sites and cyclin-dependent kinase (CDK) sites are critical for DNA repair [42,43,44]. Some examples of PIKK sites that will be of interest throughout this review are AT-mutated (ataxia telangiectasia, ATM) and AT and Rad3-related (ATR). Interestingly, RPA contains numerous PIKK sites, which may affect RPA function.

Over the decades, numerous reviews have been conducted about RPA’s structure and function [16,23,29,35,45,46,47,48,49,50,51,52,53,54,55,56,57,58,59,60,61,62]. This review will discuss novel research and more recent findings on RPA. Specifically, we will cover recent discoveries in the structure/function of RPA in DNA metabolism, interactions, and PTMs involving RPA, specific details on RPA phosphorylation in the DNA damage response (DDR) in the homologous recombination pathway, and finally conclude with a summary of necessary areas for future RPA research.

## 2. Structure of RPA

Heterotrimeric RPA has seven structured domains. RPA70 contains domains A, B, C, and F, with disordered linkers connecting the ordered domains and a zinc atom in domain C (Figure 1A). RPA32 has structured domains D and wHLH (winged helix-loop-helix), which are also connected by disordered linkers. Finally, RPA14 is entirely composed of one ordered domain named E. In Figure 1, each domain is drawn as a colored box, which indicates a known structure, whereas the black and white rectangles are disordered regions connecting the ordered, structured domains. Domains A, B, C, and D are where ssDNA binding occurs, and domains F, A, and wHLH contain sequences for protein interactions. The heterotrimeric core of RPA is formed by domains C, D, and E—one domain from each subunit. 

This unique RPA structure divides six oligosaccharide binding folds (OB-folds) among the heterotrimeric protein subunits. OB-folds are typically five-stranded closed β barrels that interact with ligands such as oligosaccharides, proteins, catalytic substrates, and more. These six OB-folds in RPA are also frequently found in ssDNA-binding proteins (Figure 1B) [53]. RPA70 contains four OB-folds that comprise the DNA binding domains within domains A, B, C, and F. RPA14 domain E and RPA32 domain D are also OB-folds [63,64,65]. The OB-folds are conserved in structure, which have more structural homology than sequence homology (Figure 1C) [53]. 

**Figure 1 genes-15-00167-f001:**
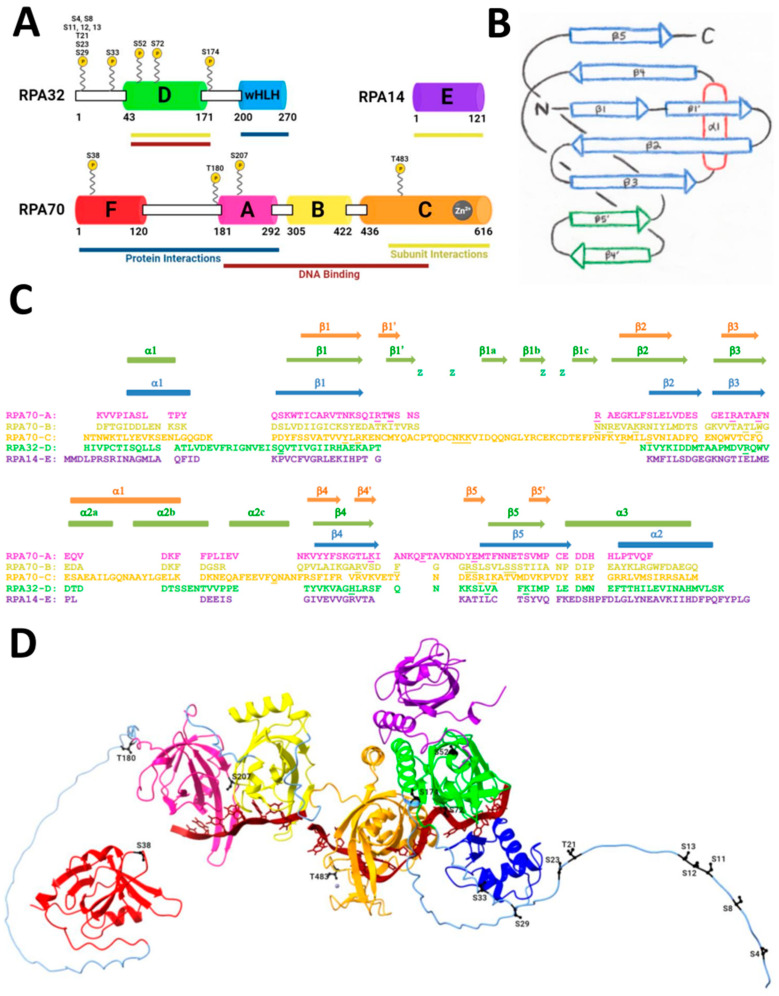
Structure of RPA. (**A**) Domain diagram of RPA heterotrimer with domain functions and phosphorylation sites. Structured domains are colored and labeled A–F and wHLH. Unstructured domains are white rectangles with a black outline. Phosphorylation sites are labeled and shown with a yellow circle indicated by P. Protein interaction domains are underlined in navy blue. DNA binding domains are underlined in maroon. Subunit interactions that form the heterotrimeric core are underlined in dark yellow. (**B**) OB-fold diagram [66]. Arrows indicate the β-strands and an oval indicates the α-helices. The blue β-strands correspond to those that comprise the OB-fold. The L12 loop lies between β1′ and β2, and the L45 loop lies between β4′ and β5′. (**C**) Sequence and secondary structure alignment of domains A–E based on structure (updated from [53]). Orange secondary structure elements represent domains A and B, green elements represent domain C, and blue elements represent domains D and E. Lowercase z indicates the Cysteine residues in domain C that bind zinc. Residues in RPA70-A, RPA70-B, RPA70-C, and RPA32-D that bind to ssDNA are underlined. RPA32-D contains putative DNA binding residues that were determined through structural alignment of human and *Ustilago maydis* RPA. (**D**) AlphaFold predicted the structure of RPA with ssDNA. Domains are colored according to the domain diagram in A. The zinc in RPA70-C is shown as a gray ball. PIKK/CDK in G2 phase phosphorylation sites are black and labeled. The phosphorylation sites: in RPA70 are S38, T180, S207, and T483; in RPA32 are S4, S8, S11, S12, S13, T21, S23, S29, S33, S52, S72, S174. ssDNA, shown in maroon, was superimposed from 4GNX [67].

Human RPA has been biochemically studied for over 30 years, with structure determination by X-ray crystallography and nuclear magnetic resonance (NMR) spectroscopy [66,68,69,70,71,72,73,74,75,76,77,78,79,80,81,82]. Unfortunately, full-length RPA has not been crystallized as a heterotrimeric protein because of the many disordered regions throughout the protein complex. By dividing RPA into purified domains, it was possible to crystallize each domain individually or with specific binding partners, such as the heterotrimeric core involving domains C, D, and E [71,75,83]. A *Ustilago maydis* structure was solved composed of ssDNA and domains A, B, C, D, and E [67]. The wHLH domain is also very loosely packed, so NMR was used to determine the wHLH structure with a bound peptide [69,82]. In 2014, Feldkamp and coworkers obtained a crystal structure of the wHLH domain without a peptide [82]. In 2015, Brosey and colleagues studied structural dynamics by NMR to evaluate RPA70-A/B domain binding to ssDNA. They obtained NMR structures for ssDNA bound to RPA70-A/B and compared their results to other structural analyses [84].

Since the flexible disorder is a significant factor in RPA structure, small angle X-ray scattering (SAXS) is another method used. This method allows for low-resolution structures from proteins in solution. In 2010, Pretto and others used SAXS specifically on RPA70 domains A and B and the N-terminal region to see how these domains bind to ssDNA. They concluded that RPA70 domains A and B are more compact in the presence of ssDNA. The N-terminal region of RPA70 did not change in structure in the presence of DNA, allowing them to conclude that there is no interaction between the two [84,85,86]. 

With these structurally known domains of RPA, AlphaFold can be handy to try to obtain a prediction of the entire RPA structure using experimentally known structures in the Protein Data Bank (Table 1). We used AlphaFold to predict various conformations of the complete protein, including the disordered regions and structured domains (Figure 1D). The modeled AlphaFold structure was used to visualize candidate PIKK/CDK Ser and Thr phosphorylation sites involved in DNA damage repair, specifically DSBs. These sites are shown in black (Figure 1D) and labeled with the corresponding amino acid and residue number. There are four phosphorylation sites on RPA70, twelve on RPA32, and none on RPA14. ssDNA was superimposed using the 4GNX structure, which interacts with RPA70 domains A, B, C, and RPA32 domain D [67]. The specific residues that bind zinc and ssDNA are indicated in Figure 1C.

## 3. The Function of RPA in DNA Double-Strand Break Repair

RPA is critical in coordinating the cell cycle with DNA replication and any DNA repair pathway involving ssDNA. The cell cycle is crucial in completing a series of events that allow cells to grow and divide and includes the following stages: interphase (I), G1 phase, S phase, G2 phase, and mitosis (M). Regulation is key to successfully replicating DNA and normal cell division within these cell cycle phases. Several laboratories have studied RPA in cell cycle regulation. RPA was determined not to be phosphorylated in the G1 phase, but RPA34 in humans and *S. cerevisiae* is phosphorylated in cells that enter the S phase and then are dephosphorylated in the M phase [45,87,88]. Various researchers have focused on RPA’s role in recruiting proteins involved in DNA repair. RPA is involved in most DNA repair pathways, including base excision repair (BER), nucleotide excision repair (NER), mismatch repair (MMR), and HR [36,89,90,91,92]. It is also important to note that HR occurs only during S and G2 phases [40,44]. 

RPA plays a critical role in HR to repair DSBs. In HR, sections of homologous sequences, typically on a sister chromatid or nearby repeated sequence, will fill in the gap left by the DSB. On each side of this DSB, a 5′ strand is resected, leaving a 3′ overhang on the DNA. This resection is done by the MRN complex (composed of RAD50, mitotic recombination 11 (MRE11), and Nijmegen breakage syndrome 1 (NBS1)), along with other proteins, including CtBP-interacting protein (CtIP), exonuclease 1 (EXO1), DNA replication helicase/nuclease DNA2, and blood syndrome RecQ like helicase (BLM). Once there is a 3′ ssDNA overhang, RPA will bind to this ssDNA and follow the canonical HR pathway that will recruit a complex that includes breast cancer type 1 susceptibility protein (BRCA1), breast cancer type 2 susceptibility protein (BRCA2), and partner and localizer of BRCA2 (PALB1). This BRCA2 complex will recruit radiation sensitivity gene 51 (RAD51) and undergo ssDNA transfer from RPA to BRCA2 complex to RAD51, and RAD51 will perform a homology search on the sister chromatid, eventually repairing the DSB. Secondly, an alternative pathway for HR includes radiation sensitivity gene 52 (RAD52). RAD52 has similar functionality to the BRCA2 complex by undergoing a ssDNA handoff from RPA and allowing RAD51 nucleation of the ssDNA again to perform a homology search on the sister chromatid, eventually repairing the DSB [40,53,93,94,95].

## 4. ssDNA and Protein Interactions involving RPA

RPA involves an assorted array of interactions with ssDNA and proteins. The function of RPA binding to ssDNA has been studied in numerous ways, for example, through atomic force microscopy (AFM), circular dichroism (CD), optical tweezers, single-molecule fluorescence resonance energy transfer (smFRET), and many more [61,96,97]. The interaction between ssDNA and RPA is a rapid but extremely stable process that is difficult to evaluate due to the binding strength. A study in 2000 concluded that the zinc-finger motif is necessary to form a stable RPA:ssDNA complex, which depends on redox reactions [98]. Others focused on the binding and unfolding of non-canonical ssDNA structures by RPA in DNA replication. The heterotrimeric core of RPA selectively binds to a G-quadruplex forming sequence. G-quadruplexes create a unique problem for RPA to unfold, and various ligands could hinder DNA replication by not allowing RPA to unfold these G-quadruplexes correctly [99].

A more recent paper described how the ssDNA binding complex can change its binding mode to allow for nucleation of RAD51. The spacing between RPA:ssDNA complexes is essential for leaving bare nucleotides in between the complexes to allow for RAD51 nucleation. The exchange of RPA for RAD51 is known to be mediated by the RAD52 middle region [100]. Overall, the RPA:ssDNA complex is essential for life and continues to be investigated.

With this insight into the RPA:ssDNA complex, multiple laboratories have investigated how RPA binds to varying lengths of nucleotides (nt), for example, 8-nt, 20-nt, or 30-nt. Kang and coworkers in 2023 investigated a protein involved in Kallmann syndrome: N-methyl-D-aspartate receptor synaptonuclear signaling and neuronal migration factor (NSMF). They determined that NSMF will colocalize and physically interact with RPA during DNA damage, allowing RPA to bind in a 30-nt binding mode. This prepares RPA to become phosphorylated by ATR, an upstream kinase involved in HR [101]. They concluded that the 30-nt binding mode of RPA enhances the phosphorylation of RPA32 by ATR, and further phosphorylation will stabilize RPA binding to ssDNA.

RPA has many protein–protein interactions involving DSB repair proteins and others (Table 2). DSB repair binding partners include: BRCA2, RAD52, RAD51, ATR/ATRIP, DNA-PKcs, DSS1, MRE11-RAD50-NBS1, p53, and PP2A [18,21,36,38,69,72,74,90,102,103,104,105,106,107,108,109,110,111,112,113,114,115,116,117,118,119,120,121,122,123,124,125,126,127,128,129,130,131,132,133,134,135,136,137,138,139,140,141,142,143,144,145,146,147,148,149,150,151,152,153,154,155,156,157,158,159,160,161,162,163,164,165,166,167,168,169,170,171,172,173,174,175,176,177,178,179,180,181,182,183,184,185,186,187,188,189,190,191,192,193,194,195,196,197,198,199,200,201,202,203,204,205,206,207,208,209,210,211,212]. The interaction of RPA with RAD52 in the alternative HR DSB repair pathway and single-strand annealing is fascinating. Without the RAD52:RPA interaction, the ssDNA handoff from RPA to RAD51 in alternative HR would not be possible, and it is also critical that RPA is phosphorylated to allow this DNA handoff [93]. Previous studies identified residues 224-271 on RPA32 and 169-326 on RPA70 that include binding sites for RAD52. It was also found that RAD52 residues 218-303 bind RPA70, along with RPA32 [21]. Although the interaction sites have been defined, structural data has yet to be available for the RPA:RAD52 complex.

BRCA2 is an essential protein involved in the canonical HR pathway. BRCA2 also completes the DNA handoff, like RAD52, by interacting with RPA. BRCA2 mutations in women can cause familial, early-onset breast cancer, so understanding these mutations and their potential impact on the interaction with RPA will become vital in understanding DNA repair, specifically in cancer cells. A study in 2003 used a cancer-predisposing BRCA2 mutation (Y42C) to investigate its interaction with RPA. They found that the Y42C mutation inhibited the interaction between RPA:BRCA2, showing that the Y42C mutation has biological importance within the human body [213]. The BRCA2 interaction with RAD51 has been studied extensively over the years, as BRCA2 promotes RAD51 nucleation on the ssDNA. Still, there has yet to be data on the interaction between BRCA2 and RPA to specifically understand how and where this interaction occurs [214].

## 5. PTMs That Regulate RPA in DNA Metabolism

PTMs are chemical modifications involved in the functional regulation of cellular proteins. RPA relies on PTMs for its proper function in the cell cycle and DNA repair, and the main PTM we will focus on in this review is phosphorylation. Phosphorylation of RPA has been studied for decades, with most studies focused on the RPA32 N-terminal region. Understanding the importance of phosphorylation on other RPA subunits in RPA’s regulation is still unresolved.

Phosphorylation occurs throughout the cell cycle, as previously described. A paper by Yates and coworkers determined two cell cycle checkpoint kinases in yeast, Mec1 and Ddc2 (ATR and ATRIP, respectively, in humans), that are essential for replication stress response and DDR. These two kinases were shown to recruit ssDNA binding to RPA through Ddc2 by phosphorylation. A yeast Ddc2-RPA structure through X-ray crystallography indicated that this interaction is necessary to mediate RPA phosphorylation during DDR [215].

An early study in 1990 by Din and colleagues determined the phosphorylation of RPA32 and RPA70 in human and yeast cells by phosphoamino acid analysis using P^32^ labeling [39]. Then, numerous labs focused on the phosphorylation of RPA because of its critical regulation in the DNA repair pathway. In 2003, Binz and others discovered that the phosphorylation of RPA32 modulates RPA-dsDNA interactions and subsequent destabilization [22]. They concluded that an intersubunit interaction between phosphorylated RPA32NT and RPA70 N-terminal domain (RPA70N) was possible. Recently, NMR and docking studies were conducted to investigate the interaction between the RPA70 N-terminal domain (RPA70N) and a phosphomimetic N-terminal peptide of RPA32 with candidate serine/threonine sites mutated to aspartic acid. The study showed the possibility that RPA32 N-terminal phosphorylation could allow for transient interaction with RPA70N, which could alter or enhance other interaction sites for proteins or DNA [216]. These conclusions indicated a possible role of RPA32 N-terminal phosphorylation binding with RPA70N to form a less disordered structure if these interactions occur in the context of the heterotrimer.

Not only is RPA involved in RPA-DNA interactions, but RPA phosphorylation also influences subcellular localization, especially when there is mitotic phosphorylation at S23 and S29 in RPA32 [217]. When the cell has a DDR, RPA becomes hyperphosphorylated [50,56]. For example, in response to DSBs, RPA is a substrate of ATM kinase that phosphorylates RPA32 at Thr21 and most likely others [43]. In 2014, a study was performed on RPA hyperphosphorylation involving the HR pathway in cell cycle phases where HR DSB repair occurs. In these studies, a human squamous cell carcinoma cell line (UM-SCC-38) was synchronized in the S and G2 phases [218,219,220]. Phosphorylation sites in S and G2 phases with and without DSBs were analyzed with western blots using all available antibodies for phosphorylation sites on RPA32. In the G2 phase, chromatin-bound immunoprecipitation phosphorylation occurred at S4/8, S12, T21, S23, and S33 of RPA32 after DSBs. 

A global view of RPA heterotrimer phosphorylation was explained using a high-quality RPA70 antibody and isoelectric focusing (IEF) to separate RPA isoforms based on the number of phosphorylations present. Capillary IEF immunoassay data of the phosphorylated RPA heterotrimeric isoforms with and without DSB treatment, synchronized in the G2 phase of the cell cycle, were collected (Figure 2) [40]. In this study, data showed that RPA is always a phosphoprotein in the G2 phase by having up to seven phosphates (Figure 2A, blue line). Hyperphosphorylated isoforms generated after DSBs contained up to 13 or 14 phosphates. IEF identified some of these sites with phosphospecific antibodies (Figure 2B–E). The hyperphosphorylated forms with 13–14 phosphates included pS4/8, pT21, and pS33. Several of the sites had yet to be characterized before. DSBs showed increased phosphorylation by PIKK and CDK kinases in S and G2 phases, providing new information on candidate PIKK and CDK sites for future research.

This research provided pertinent information about RPA heterotrimeric phosphorylation in response to DSBs in the S and G2 phases of the human cell cycle. Several of these phosphorylation sites are PIKK-specific sites in the DNA binding and protein interaction domains of RPA. A summary figure describes possible phosphorylation sites before and after DSBs (Figure 3). This study provided data about phosphatase and kinase action in remodeling RPA isoforms in response to DNA damage [40].

## 6. Future Directions for RPA Research

As previously described, PTMs, such as phosphorylation, play a crucial role in RPA regulation. Future RPA research should focus on the many isoforms of phosphorylated RPA and the complicated phosphorylation patterns of RPA to determine which ones are necessary for DNA repair, cell cycle regulation, or other metabolism pathways. Phosphorylation is critical in RPA regulation and impacts some DNA repair pathways.

RPA phosphorylation could impact protein–protein interactions in DNA repair, DNA DSB repair, replication, gene expression, and more (Table 2). These interactions could be explored through various binding studies, including thermal shift assays, biolayer interferometry (BLI), and surface plasmon resonance (SPR). CD spectroscopy could show if phosphorylation changes secondary structure [221]. In vivo and in vitro activity assays would be beneficial to determine where these phosphorylation sites impact RPA function and interactions with proteins. Phosphomimetics, for example, serine/threonine to glutamic acid mutations, could be used in these problematic studies. Understanding how phosphorylation affects the RPA–protein interactions related to DNA DSB repair would give insight into how to target the interactions only involved in cancerous cells.

Understanding the role of RPA phosphorylation could lead to the creation of drugs that specifically target these forms of protein and complexes. In cancer cells deficient in the BRCA2 complex, the alternative HR pathway involving RAD52 must step up to continue HR, but most importantly, it must continue to repair the broken DNA. Deng and coworkers demonstrated that RPA phosphorylation significantly stabilized the RPA:RAD52 complex and regulates strand transfer [93]. So, the phosphoRPA:RAD52 protein complex would be a potential target for a therapeutic inhibitor of complex formation. With this interaction inhibited in HR-deficient cancer cells, such as high-grade serous carcinoma ovarian cancer, these cells will not be able to survive. The most significant outcome of these potential pharmaceuticals will be the lack of impact on the non-cancerous cells in the body with functional canonical BRCA2 pathway.

Visualization of proteins is key for adequately understanding how they function, and the disordered regions of RPA make this process difficult. It is known that RPA alone and the RPA:RAD52 complex (with and without ssDNA) cannot be crystallized due to this intrinsic disorder. Techniques such as cryo-electron microscopy (cryo-EM) will be a helpful tool to structurally analyze RPA in its entirety [83,222,223]. Utilizing cryo-EM to study RPA protein–protein interactions (Table 2) would allow for investigation and fill a massive gap in the knowledge of these interactions. 

## 7. Conclusions

RPA is a heterotrimeric protein involved in cell cycle regulation, repair, and countless other pathways involved in DNA metabolism. Regulation of this multifarious protein is essential in DNA metabolism. The structure of RPA has been studied for several decades, with only a handful of experiments on the entire protein because of the disordered regions. AlphaFold has predicted RPA as a complete heterotrimeric protein, but the disordered regions vary in each prediction. The model has allowed us to visualize where the phosphorylated PIKK/CDK sites are throughout the protein. This lets us understand which interactions in DNA binding, protein–protein interactions, or formation of the heterotrimeric core might be altered. 

Regulation of RPA is a key component involved in various processes, such as PTMs. PTMs like acetylation, ubiquitination, SUMOylation, ADP ribosylation, and phosphorylation are fundamental to RPA regulation. Throughout this review, the main PTM focused on phosphorylation because of its complexity and misunderstanding throughout the years. The number of phosphorylation combinations is mind-boggling. Phosphorylation of RPA is complex and an important part of controlling multiple interactions, maintaining cell cycle regulation, and repairing DNA. Therefore, RPA is an intriguing mystery that needs in-depth characterization of phosphorylation in DNA double-strand break repair to understand RPA functions fully.

## Figures and Tables

**Figure 2 genes-15-00167-f002:**
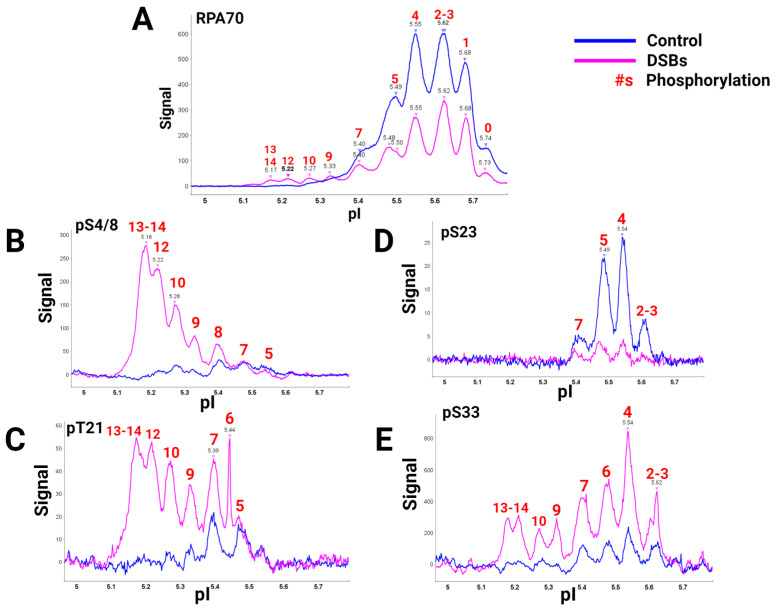
RPA isoforms from cells synced in the G2 phase either with DNA damage (blue) or without DNA damage (pink), separated with isoelectric focusing using a pH gradient from 5-6 and probed with phospho-specific antibodies. RPA heterotrimer probed with (**A**) anti-RPA70-CT, (**B**) phosphoRPA32(S4/8), (**C**) phosphoRPA32(T21), (**D**) phosphoRPA32(S23), and (**E**) phosphoRPA32(S33) antibodies. The red number above the corresponding peaks indicates the number of phosphorylations on each isoform. Figure adapted from [40].

**Figure 3 genes-15-00167-f003:**
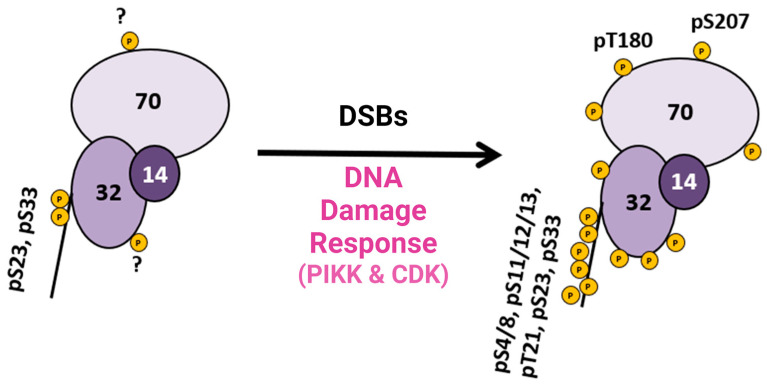
Summary diagram of RPA phosphorylation before and after DSBs. RPA is always phosphorylated (some sites are unknown and indicated with a ?), but after DDR is activated, RPA becomes hyperphosphorylated by the PIKK family of kinases and CDK cell cycle kinases, which phosphorylate RPA. Phosphorylation sites after DNA damage are shown in Figure 1D. Adapted from [40].

**Table 1 genes-15-00167-t001:** Human RPA structures in the Protein Data Bank (PDB).

Subunits/Domain	Experiment Information	PDB ID	Resolution (Å)	Citation
	X-ray; MRE11; RPA70(1-121)	8K00	1.4	[76]
RPA70-F	X-ray; RAD9; RPA70(1-120)	8JZY	1.5	[76]
	X-ray; ETAA1; RPA70(1-120)	8JZV	1.5	[76]
	X-ray; ATRIP; RPA70(1-123)	7XV4	1.6	[76]
	X-ray; HelB; RPA70(1-120)	7XV1	1.8	[76]
	X-ray; BLMp1; RPA70(1-120)	7XV0	1.5	[76]
	X-ray; RMI1; RPA70(1-120)	7XUV	1.6	[76]
	X-ray; WRN; RPA70(1-120)	7XUT	1.6	[76]
	X-ray; BLMp2; RPA70(1-120)	7XUW	1.8	[76]
	X-ray; drug; RPA70(1-123)	5E7N	1.21	[77]
	X-ray; Dna2 peptide; RPA70(1-123)	5EAY	1.55	[70]
	X-ray; PrimPol(514-525); RPA70(1-123)	5N85	2	[74]
	X-ray; PrimPol(480-560); RPA70(1-123)	5N8A	1.28	[74]
	X-ray; drug; RPA70(1-123)	4R4T	1.28	[78]
	X-ray; drug; RPA70(1-123)	4R4Q	1.35	[78]
	X-ray; drug; RPA70(1-123)	4R4O	1.33	[78]
	X-ray; drug; RPA70(1-123)	4R4I	1.4	[78]
	X-ray; drug; RPA70(1-123)	4R4C	1.4	[78]
	X-ray; drug; RPA70(1-123)	4O0A	1.2	[73]
	X-ray; drug; RPA70(1-123)	4LWC	1.61	[73]
	X-ray; drug; RPA70(1-123)	4LW1	1.631	[73]
	X-ray; drug; RPA70(1-123)	4LUZ	1.9	[73]
	X-ray; drug; RPA70(1-123)	4LUV	1.4	[73]
	X-ray; drug; RPA70(1-123)	4LUO	1.54	[73]
	X-ray; drug; RPA70(1-123)	4IPH	1.94	[79]
	X-ray; drug; RPA70(1-123)	4IPG	1.58	[79]
	X-ray; drug; RPA70(1-123)	4IPD	1.51	[79]
	X-ray; drug; RPA70(1-123)	4IPC	1.22	[79]
	X-ray; drug; RPA70(1-123)	4IJL	1.7	[80]
	X-ray; drug; RPA70(1-123)	4IJH	1.498	[80]
	X-ray; p53N(33-60)/RPA70(1-123)	2B3G	1.6	[72]
	X-ray; RPA70(1-123)	2B29	1.6	[72]
	NMR; RPA70(1-114)	1EWI	-	[68]
RPA70-A/B	X-ray; RPA70(181-422)	1FGU	2.5	[63]
	X-ray; ssDNA; RPA70(181-422)	1JMC	2.4	[66]
	NMR; XPA-MBD/RPA70(168-326)	1D4U	-	[20]
RPA32-wHLH	NMR; UNG2(73-88); RPA32(171-270)	1DPU	-	[69]
	X-ray; RPA32(197-270)	4OU0	1.4	[82]
RPA32/RPA14	X-ray; full length	2Z6K	3	[75]
	X-ray; full length	2PQA	2.5	[75]
	X-ray; full length	2PI2	2	[75]
	X-ray; RPA32(43-172)/RPA14(1-121)	1QUQ	2.5	[81]
RPA70/RPA32/RPA14	X-ray; Zn^2+^; RPA70(435-616)/RPA32(43-171)/RPA14(1-121)	1L1O	2.8	[71]

**Table 2 genes-15-00167-t002:** Human proteins that interact with RPA.

Interacting Protein	Interaction Site on RPA	DNA Metabolism Pathway	Citation
AID	RPA32	Immunoglobulin diversification	[107]
* Ajuba	RPA70	DNA damage response (DDR)	[108,109]
** ATR/ATRIP	RPA70-F	Checkpoint signaling, DNA repair	[110,111,112]
BID	RPA70-F	Replication stress response	[113]
* BLM	RPA70-A/B	DNA unwinding, resection	[114,115,125]
** BRCA2	?	Homologous Recombination (HR)	[105]
DDX11	?	Chromosome segregation	[193]
** DNA-PKcs	?	DNA repair	[18,116]
** DSS1	RPA70	HR	[121]
* ETAA1	RPA70-F/RPA32	ATR activation, repair at stalled replication forks	[117,118,119,120]
* EXO5	RPA70-F	Intrastrand crosslink repair	[181]
* FANCJ	RPA70	DNA repair, genome stability	[123]
* FBH1	RPA32	DNA unwinding, resection	[193]
HELB	RPA70-F	Replication stress response	[179,180]
HERC2	RPA70	Replication	[124,125]
HIRA	RPA70-C	Chromatin remodeling	[126]
Histones H3 and H4	RPA70-F	Chromatin remodeling	[127]
* HLTF	RPA70	Genome stability	[195,196]
HSF1	RPA70	Gene expression	[122]
* Menin	RPA32	Genome stability	[129,130]
** MRE11-RAD50-NBS1	RPA70-F	HR	[131,132]
Nucleolin	RPA14	Replication (stress)	[133,134,135]
NSMF	RPA32	DDR	[101]
** p53	RPA70-F	HR	[136,137,138,139,140]
* p53BP1	RPA70/RPA32	DDR	[106]
* PALB2	RPA32	Recovery of stalled replication forks	[141]
Papillomavirus E1	RPA70-A	Replication	[182,183]
Parvovirus NS1	RPA70/RPA32	DNA unwinding, resection	[187]
PCNA	RPA70	Replication	[191]
Polδ	RPA70	Replication	[188]
Pol-Prim	RPA70-F/A/B	Replication restart, DNA damage tolerance	[102,203]
** PP2A	RPA32	DDR	[204]
* PRP19/BCAS2	RPA70-F/C	DNA repair	[144]
* PTEN	RPA70	Genome stability	[145]
* RAD9/RAD1/HUS1 (9-1-1)	RPA70/RPA32	DDR	[146]
RAD17	RPA70-F	DDR, replication stress response	[147,178,205]
** RAD51	RPA70-A	Recombination	[103,150,151,206]
** RAD52	RPA70-A/B & RPA32-wHLH	DNA repair	[21,69,152,153,154,155,156,207]
RECQL1	RPA70	DNA unwinding	[198,199]
RECQ5β	?	DNA unwinding	[200,201]
RFC	RPA70	DNA unwinding	[188,189]
* RFWD3	RPA32-wHLH	DNA repair	[157,158,195]
* RNaseH	?	Transcription, DNA repair	[159]
* RNF4	RPA70	DNA DSB repair by HR	[202]
SENP6	RPA70	Unperturbed DNA replication	[38]
SMARCAL1/HARP	RPA32-wHLH	Replication fork restart	[160,161,162,208]
SV40 Large T antigen	RPA70-A/B & RPA32-wHLH	Replication	[182,183,184,185,186]
* Tipin	RPA32-wHLH	DDR	[164,165]
* UDG	RPA32-wHLH	Base excision repair	[69,166,190]
* UNG2	RPA32-wHLH	Base excision repair	[69,166]
* WRN	RPA70-A/B	DNA unwinding, resection	[115,170]
* XPA	RPA70-A & RPA32-wHLH	Nucleotide excision repair (NER)	[69,166,171,209,210,211]
* XPF-ERCC1	RPA70	NER	[174,175,176]
* XPG	?	NER	[174,176,212]

* Involved in DNA repair; ** Involved in DNA DSB repair.

## Data Availability

Not applicable.

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
