# Peer review of "The Intriguing Mystery of RPA Phosphorylation in DNA Double-Strand Break Repair"

_genes, 2024, doi:10.3390/genes15020167_

Round 1
Reviewer 1 Report
Comments and Suggestions for Authors
Replication protein A (RPA) is a single strand DNA binding protein, composing of three subunits of 70, 32, and 14 kDa. Because RPA is involved in almost every aspect of DNA metabolism in eukaryotes, many work have been dedicated to RPA.
Dr. Gloria Borgstahl’s lab has been working on RPA for decades, mainly from its structural perspective. In the current review, Fousek-Schuller and Borgstahl tried to summarize recent understanding of RPA phosphorylation regulation in cell cycle.
As author mentioned: numerous reviews about RPA structure and function [16, 23, 29, 35, 45-61], a key question is how this review is different from and better than these published? This is also my biggest concern.
Here are some concerns:
1. I have to admit it is a challenging job to review RPA: what to cover and what not to cover in terms of its functions, how deep and surface in a given topic, and how old and how new (relatively easy) in terms of literature. It seems to me authors wanted to cover RPA phosphorylation in cell cycle as authors mentioned two times in the abstract; however, authors wrote “specific details on RPA phosphorylation in the repair of DSBs” (line 60-61) and Figures 3 and 4 are dedicated to DNA damage, so please tell readers clearly what you will focus in this review?
2. The title is too “mystery” or too broad (related to Q1). Because DNA metabolism includes replication, recombination, repair…and RPA can be regulated by PTMs, protein-protein interactions, protein-ssDNA interaction…while the current review is mainly focusing on phosphorylation of RPA in cell cycle (in fact DNA repair as well), so might specify the title.
3. As my understanding authors wanted to cover RPA’s roles in DNA replication (or called cell cycle) and DNA repair, I feel the cell cycle section is weak.
4. I am sure authors know the difference between cell cycle and DNA replication, as DNA replication just happens in S phase, and cell cycle can be regulated in the rest 3 phases and many more molecules are involved, so they are not interchangeable.
5. Figure 4 is kind of duplicated with Figures 1 and 2. To stand differently, their kinase might be included.
6. Using a table to show RPA-interacting proteins?
7. Combine Figure 1 and 2? Since they are in the matched color, it will be good to put them together.
8. In keywords, cell cycle or DNA metabolisms should be included.
9. Figure 2 title, wtRPA, anywhere you have discussed RPA mutations?
10. Figure 3 and 4 are from author’s 2014 paper, need citation in caption.
11. The paper Yates et al. “A DNA damage-induced phosphorylation circuit enhances Mec1ATR Ddc2ATRIP recruitment to Replication Protein A”, PNAS 2023 might be cited.
Reviewer 2 Report
Comments and Suggestions for Authors
This review on RPA reads very well and is a nice addition to the field. The Alpha Fold model of RPA shown in Fig 2 is useful. It might be better if this was combined with the domain structure in Fig. 1, and if the domains were labeled in Fig2. Also, some presentation, perhaps in a table, of the experimental structures that have been determined in the context of Fig. 2, would be helpful. In particular, overlaying of bound ssDNA from any experimental structures onto Fig. 2 would be helpful, so that the reader can see how ssDNA would be bound.
The authors have focused the PTMs on phosphorylation, as that relates most closely to their contributions. The review is not comprehensive in that sense (same for protein-protein interactions), but that is fine.
Round 2
Reviewer 1 Report
Comments and Suggestions for Authors
Authors have addressed most concerns.